# The Impact Path of Digital Literacy on Farmers' Entrepreneurial Performance: Based on Survey Data in Jiangsu Province

## Shiyu Ji * and Jincai Zhuang

School of Management, Jiangsu University, Zhenjiang 212013, China; 15195120672@163.com

* Correspondence: jsy19990714@163.com

**Abstract:** In the era of digital economy, digital literacy plays an important role in the process of enabling farmers' entrepreneurship with digital technology. However, there are few studies in the literature on the impact and mechanism of digital literacy on farmers' entrepreneurial performance. From the perspective of digital literacy, combining high-level theory and resource-based theory, this paper constructed a chain intermediary model of "digital literacy–entrepreneurial bricolage–entrepreneurial opportunity identification–entrepreneurial performance" based on the logical chain of "resource–opportunity–performance". A hierarchical regression analysis and bootstrap method were used to analyze 308 samples of effective entrepreneurial farmers. The results show the following: (1) Digital literacy has a significant positive impact on the entrepreneurial performance of rural households. (2) Entrepreneurial bricolage and entrepreneurial opportunity identification play an intermediary role between rural households' digital literacy and entrepreneurial performance. (3) There is a chain mediating effect of "digital literacy–entrepreneurial bricolage–entrepreneurial opportunity identification–entrepreneurial performance". The research conclusions can broaden the research on the pre-influencing factors of farmers' entrepreneurial performance and the driving effect of digital literacy and provide enlightenment on how to promote the improvement of farmers' entrepreneurial performance and rural social and economic sustainable development.

**Keywords:** digital literacy; farmers' entrepreneurial performance; entrepreneurial bricolage; entrepreneurial opportunity identification; mediating effect

## 1. Introduction

The report to the 20th National Congress of the Communist Party of China put forward the strategy of "implementing the strategy of giving priority to employment", emphasizing "establishing a mechanism for promoting entrepreneurship and creating jobs through multiple channels and flexible employment, coordinating urban and rural employment policies, and actively guiding rural labor to find jobs". In recent years, entrepreneurship has been regarded as a key factor in promoting sustainable economic development due to its role in expanding employment and optimizing the industrial structure [1]. Therefore, the CPC Central Committee and The State Council attach great importance to innovation and entrepreneurship, especially in the field of rural revitalization. Policy documents such as the Strategic Plan for Rural Revitalization (2018–2022) and the Opinions on Promoting High-quality Development of Business Start-ups in Rural Areas (No. 104, 2020) have been issued in succession, greatly increasing farmers' enthusiasm for entrepreneurship and driving the flow and convergence of production factors such as technology, talent and capital to rural areas. By March 2022, more than 11.2 million people had returned to their hometowns to start businesses, and remarkable progress had been made in rural innovation and entrepreneurship. However, entrepreneurship is a process in which entrepreneurs identify entrepreneurial opportunities and integrate and utilize resources to achieve opportunity value creation [2]. As a special entrepreneurial group, rural households have distinct internal limitations and external constraints, which are mainly reflected in blocked information sources, severe constraints on entrepreneurial resources, and insufficient innovation

ability [3]. The phenomenon of low entrepreneurial quality and low entrepreneurial performance is extremely serious. Therefore, how to help farmers improve their entrepreneurial performance has become the focus of research on rural entrepreneurship under the background of the implementation of rural revitalization strategy.

With the rapid development of the digital economy in China, new digital information technology represented by the Internet rises vigorously and gradually becomes the endogenous power of rural economic development [4]. For rural households with low entrepreneurial performance, digital economy promotes the increase in entrepreneurial benefits of rural households to a considerable extent by broadening information channels, expanding production and sales modes, and improving financing channels [5]. It is worth noting that due to the existence of a digital divide, not all entrepreneurial farmers can enjoy digital empowerment in the "reservoir" of the digital economy. According to the Survey and Analysis Report of China's Rural Digital Literacy under the Background of Rural Revitalization Strategy, the Internet penetration rate in rural China reached 57.6% in 2021, and the gap between urban and rural digital infrastructure narrowed significantly, while the score of rural digital literacy was only 18.6, lower than the average of 57%. It can be seen that the current digital divide problem has shifted from the access gap to the digital literacy gap. However, relevant studies have found that having a certain degree of digital literacy is a prerequisite for individuals to integrate into the digital society, enjoy digital dividends, and open up space for entrepreneurship, employment, and income increase, and farmers are no exception [6]. In other words, the digital literacy underlying the accessibility of digital technology is the key to influencing the entrepreneurial behavior and performance of rural households in the period of "digital revolution". According to this logic, the digital literacy of rural households is closely related to entrepreneurial performance. In what way does digital literacy affect the entrepreneurial performance of rural households? And how can rural households' digital literacy be improved to improve their entrepreneurial performance? Exploring this problem is helpful to effectively cultivate rural households' digital literacy, help rural households master and apply digital technologies and skills in the process of entrepreneurship, and thus promote the improvement of rural households' entrepreneurial performance.

As digital literacy plays an increasingly important role in farmers' production and life, relevant scholars have conducted research on its driving effect. For example, Lanlan and Yanling (2022) empirically tested the positive impact of farmers' digital literacy on their overall participation in rural digital governance [7]. Honggen et al. (2022) believe that digital literacy can help improve rural residents' willingness to classify household garbage and encourage them to actively practice garbage classification behavior [8]. From the perspective of common prosperity, Depeng et al. (2022) found that digital literacy can promote the accumulation of farmers' property income [9]. In addition, some studies have indirectly investigated the impact of digital literacy and peasant household entrepreneurship. Jie et al. (2022) discussed that the promotion of digital literacy can help improve the entrepreneurial activities of rural residents and alleviate multidimensional relative poverty [10]. Dunping et al. (2022) believe that returning home to start businesses can alleviate the multidimensional relative poverty of rural households by improving their digital literacy [11]. Xiaojing et al. (2022) found that digital literacy has a positive impact on farmers' entrepreneurial behavior, and such an impact has a significant positive spatial spillover effect [12]. However, current academic discussions on digital literacy mostly focus on rural governance, multidimensional poverty of rural households, etc., and the role of digital literacy in the field of entrepreneurship has not been deeply discussed. Even though a few empirical studies confirm that digital literacy has a positive effect on rural households' entrepreneurial behaviors, however, it remains to be further explored how to influence the entrepreneurial performance of farmers.

In view of the above problems and the lack of research, this paper uses the micro-survey data of farmers in Jiangsu Province in 2022 to investigate the digital literacy of entrepreneurial farmers from three dimensions: digital cognitive identification, digital eval-

uation application, and digital communication sharing, based on the basic connotation of farmers' production and living conditions and literacy. Centering on the core issue of how digital literacy promotes the improvement of rural household entrepreneurial performance, this paper introduces entrepreneurial bricolage and entrepreneurial opportunity identification into the research framework according to the logical chain of "resource–opportunity–performance" and explores the mediating and chain-mediating roles of the two in the relationship between rural household digital literacy and entrepreneurial performance, in order to reveal the intermediate transformation path of digital literacy affecting rural household entrepreneurial performance. To enrich the literature in the field of the digital literacy driving effect and improve the existing research results on entrepreneurial performance, it also inspires the government and entrepreneurial farmers to pay attention to the cultivation of digital literacy and provides theoretical references and practical suggestions for entrepreneurial farmers to better realize digital technology-enabled entrepreneurship.

## 2. Theoretical Analysis and Research Hypothesis

### 2.1. Digital Literacy and Peasant Household Entrepreneurial Performance

Based on the high-level theory, when facing the organizational situation, entrepreneurs will exert influence on the resources, opportunities, and strategies of the enterprise according to their personal characteristics within their limited vision, thus affecting the outcome and output of the enterprise [13]. Digital literacy refers to the attitude and ability of an individual to correctly and reasonably use information tools, reasonably use digital resources, and effectively communicate with others (Famin, 2021) [14]. As a personal trait of entrepreneurs, digital literacy can affect entrepreneurial performance. The specific theoretical framework model is shown in Figure 1. First of all, the market radius of traditional farmers' entrepreneurship is small, so the products or services provided by them have strong homogeneity, which leads to low quality and poor performance of entrepreneurship [15]. With the development of digital economy, entrepreneurial farmers can use digital platforms to break market limitations and meet a wider range of market demands. However, digital literacy affects whether individuals use digital information tools and their ability to use them. Therefore, farmers with high digital literacy are willing and skilled to take advantage of online platforms such as e-commerce and digital payment to improve the convenience of participating in market competition and the digital marketing level, such as selling products through common methods such as Douyin and video delivery, so as to achieve low cost and expand sales channels. The original market radius can be broken to solve the spatial restriction of product market and the problem of product homogeneity and promote the increase in farmers' sales income [10]. At the same time, the improvement of digital literacy can help farmers to understand the needs of consumers and other digital information. Farmers with high digital literacy can use the instant communication function of the platform to maintain smooth communication with the trading objects of products or services and form a better trading relationship, so as to help entrepreneurial farmers expand the trading scale and gain competitive advantages in providing products and services that customers want, and finally effectively promote the improvement of entrepreneurial performance.

Second, due to the long-term influence of the urban–rural dual system, the rural education level is low, the previous experience of most entrepreneurial farmers is mainly concentrated in the field of agricultural production, and their enterprise management experience is relatively short [16]. Although some rural households have received the edification of urban industrial civilization through urban migrant work, due to the low level of standardized and systematic education, the position of rural households in urban migrant work is relatively low, and they lack the opportunity to learn the knowledge and skills of modern enterprise management. At the same time, entrepreneurial farmers have a homogeneous social network, and it is difficult for them to obtain information about business operation and management from their relatives and friends, which inhibits the improvement of the entrepreneurial farmers' operation and management ability and leads

to poor entrepreneurial performance. Relevant studies show that the farmers' knowledge, skills, and management ability are crucial to the improvement of entrepreneurial performance, and entrepreneurial learning can affect the improvement of entrepreneurial ability. If entrepreneurial farmers can imitate and learn from the external environment, they can overcome the knowledge and skills barriers required for entrepreneurship [17]. Nowadays, digital information technology broadens information access channels and social networks, which makes it possible for entrepreneurial farmers to overcome the barriers of knowledge ability, while digital literacy affects the process of farmers using digital information technology to overcome the barriers of management ability. Digital literacy has obvious human capital accumulation and peer effects. Farmers with high digital literacy can use "dry learning" to accumulate human capital, which is conducive to improving the level of human capital of farmers, strengthening their operation and management abilities, as well as social interaction, thus promoting the growth of their productivity and stimulating the vitality of innovation and entrepreneurship [18]. At the same time, digital literacy can affect farmers' attitudes and digital communication and sharing abilities. Farmers with high digital literacy can exchange entrepreneurial experiences, learn, and share relevant knowledge with other entrepreneurs through digital channels through knowledge spillover and the sharing effect of social networks, so as to rapidly improve their management ability and stimulate innovative behavior and then promote the improvement of entrepreneurial performance [19].

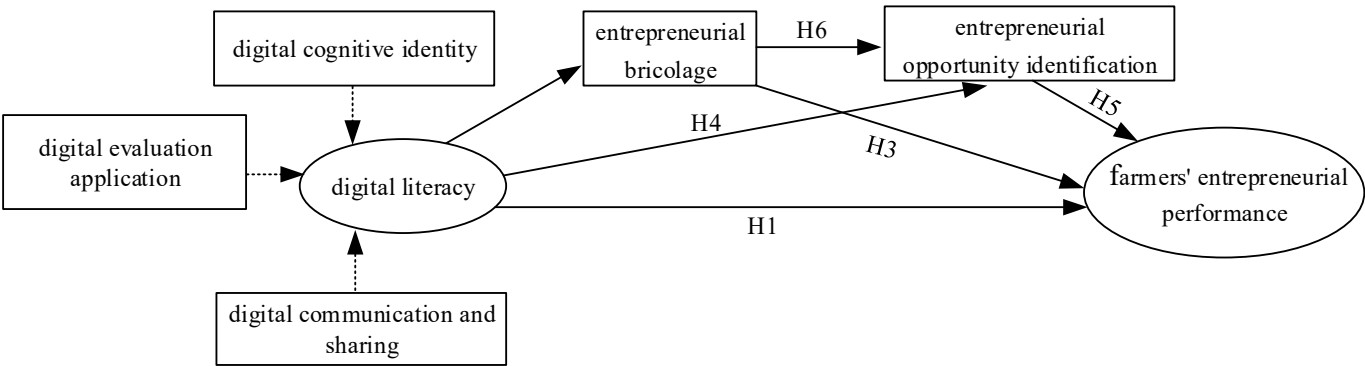

**Figure 1.** Theoretical framework model.

**H1:** *Digital literacy has a positive impact on the entrepreneurial performance of rural households.*

*2.2. Intermediary Role of Entrepreneurial Bricolage*

The entrepreneurial bricolage theory emphasizes that entrepreneurs examine the value and function of existing or redundant resources and use resource reorganization to solve the impacts of the amount of resources on the performance of their enterprises, which has important inspirational value for farmers with limited entrepreneurial resources and low entrepreneurial performance [20]. Entrepreneurial farmers are faced with resource dilemmas such as weak human and social capital and backward infrastructure. As digital information technology enables farmers to start businesses, their digital literacy can promote the farmers' use of digital technology within limited resource horizons to piece together fragmented and easily ignored resources, make innovative use of existing resources, eliminate resource constraints, and create highly valuable heterogeneous resources. Thus, it can help farmers obtain resources and competitive advantages and finally bring about changes in entrepreneurial performance [21]. First of all, currently rural areas still maintain the traditional cultural characteristics of collectivism in China, so farmers are more willing to rely on relatives, blood ties, geographical relatives, and villagers in the process of entrepreneurial piecing together, which leads to the closure of their social circle and serious homogeneity of resources [22]. However, farmers with high digital literacy have a higher

attitude and ability to apply digital evaluation. They can not only stabilize social network relationships of acquaintances but also obtain redundant resources provided by strong relationships. In addition, they can also skillfully apply diversified digital tools, frequently carry out remote interpersonal interaction, break the social circle of acquaintances, and then expand their social network and obtain more abundant weak relationship resources. The interaction between strength and weakness brought by digital literacy helps farmers collect various fragmented, disordered, and idle resources and "make do" and "reconstruct" the resources embedded in the social network through entrepreneurial bricolage, so as to achieve orderly and centralized utilization, so as to help entrepreneurial farmers reduce resource acquisition costs, enhance their own competitive advantages, and improve entrepreneurial performance. Second, entrepreneurial bricolage is actually a process of continuous learning. If farmers have high digital literacy, they have a strong learning attitude and ability toward digital technology and rich digital knowledge, so they can proficiently learn and master complex key knowledge quickly from massive materials and information on digital platforms [4]. This helps farmers, when faced with resource scarcity, make use of such knowledge and experience to analyze seemingly redundant resources at hand from a comprehensive and professional perspective, such as resource advantages, attributes, and connections among various resources, so as to piece together fragmented and easily ignored redundant resources with less time cost and a lower economic cost. They can break through the limitations of existing resources to carry out utilization-type innovation, create high-value heterogeneous resources to a certain extent, and ultimately bring changes in entrepreneurial performance [23]. Based on the above analysis, this paper proposes the following hypotheses:

**H2:** *Digital literacy positively affects farmers' entrepreneurial bricolage behavior.*

**H3:** *Entrepreneurial bricolage plays an intermediary role in the relationship between digital literacy and rural household entrepreneurial performance.*

### 2.3. Intermediary Role of Entrepreneurial Opportunity Identification

According to the high-level theory, the differences in personal characteristics of entrepreneurs will significantly affect their understanding of reality, thus influencing the decision-making of entrepreneurial behavior and ultimately affecting the performance of organizations [24]. Entrepreneurship is a process in which entrepreneurs identify entrepreneurial opportunities and integrate and utilize resources to create opportunity value. Therefore, whether farmers can correctly identify and evaluate entrepreneurial opportunities is the starting point and key factor for their entrepreneurial success [25]. Rural households' digital literacy can promote their identification and development of entrepreneurial opportunities and timely adjustment of business strategies and ultimately bring changes in entrepreneurial performance. First of all, entrepreneurial farmers generally have low educational levels and have been engaged in traditional agricultural production for a long time, so their management experience and knowledge reserve are insufficient, which restricts farmers' grasp of entrepreneurial opportunities [26]. With the rapid development of digital technology, enterprises are driven by digital technology to break the equilibrium of the original market, and the unbalanced market has produced a large number of entrepreneurial opportunities. Farmers with high digital literacy have keen digital awareness, rich digital knowledge, and high digital skills. They are willing to use digital tools to contact and learn knowledge and skills in modern enterprise operation and management, improve their management experience and knowledge reserve, and solve the ability dilemma. This helps farmers to accurately identify and develop hidden entrepreneurial opportunities in unbalanced markets, thus driving the improvement of entrepreneurial performance [12]. Second, the identification of entrepreneurial opportunities is the result of the rational collection and acquisition of information, which is spread and shared through social networks [27]. The existence of the urban–rural dual system leads to the break between market information

and farmers. However, farmers' information sources mainly rely on an acquaintance society, and the information has strong homogeneity. Moreover, farmers' ability to collect and screen market information is relatively low. This leads to farmers finding it difficult to grasp the market information in time, missing the opportunity to develop at the appropriate time. As digital information technologies help farmers grasp market opportunities, digital literacy is increasingly important for identifying entrepreneurial opportunities. Farmers with high digital literacy use various network communication platforms with higher frequency and efficiency and can timely and accurately capture market dynamics and policy information changes through network information channels. At the same time, they expand social networks and enhance social interaction in other individual communication processes. In order to obtain professional advice and guidance, expand heterogeneous knowledge, and timely exchange and acquisition of tacit information. This not only helps farmers to have access to more and higher quality start-up market information but also reduces the cost of information acquisition, so as to help farmers to identify and evaluate entrepreneurial opportunities in a timely manner and ultimately improve entrepreneurial performance [28]. Based on the above analysis, this paper proposes the following hypotheses:

**H4:** *Digital literacy positively affects the identification of entrepreneurial opportunities for farmers.*

**H5:** *Entrepreneurial opportunity identification plays an intermediary role in the relationship between digital literacy and entrepreneurial performance of rural households.*

*2.4. The Chain Mediating Effect of Entrepreneurial Bricolage and Entrepreneurial Opportunity Identification*

According to the resource-based theory and Timmons Entrepreneurial Factors Model theory, the availability of resources and the richness of entrepreneurial opportunities possessed by an enterprise affect its entrepreneurial activities to a considerable extent. As mentioned above, digital literacy, as a personal trait, will have an impact on farmers' entrepreneurial bricolage behavior, which in turn helps farmers identify and develop new opportunities through the reallocation of resources at hand and ultimately promotes the improvement of entrepreneurial performance. Farmers with high digital literacy are better at resource bricolage, so they are more likely to re-understand and creatively utilize the resources, knowledge, and experience at hand in the context of resource shortage, thus obtaining a greater set of entrepreneurial opportunities. Farmers will make judgments based on such experience and knowledge and adjust their business strategies and models in time according to market demands, so as to create and identify more profitable entrepreneurial opportunities and ultimately improve entrepreneurial performance [29]. In addition, identifying or creating market opportunities is the starting point of farmers' entrepreneurship. For the vast majority of farmers, without relatively stable and sufficient capital and other resources, they will hardly choose the entrepreneurial model with high returns, thus leaving them unable to identify and develop market opportunities [30]. Farmers with high digital literacy can skillfully apply digital information technology to resource bricolage, and resource bricolage can help farmers break traditional inertia thinking, make full use of non-standardized resources for the creative bricolage of resources, and further improve their entrepreneurial performance in the process of promoting farmers' opportunity identification and development. According to the above theoretical logic, it can be inferred that digital literacy will be conducive to the identification of entrepreneurial opportunities in the process of affecting the entrepreneurial bricolage of farmers and thus affect the entrepreneurial performance of farmers. Based on the above analysis, this paper proposes the following hypotheses:

**H6:** *Entrepreneurial bricolage and entrepreneurial opportunity identification play a chain intermediary role between digital literacy and peasant household entrepreneurial performance, that is, digital literacy promotes the identification of entrepreneurial opportunities by influencing*

*the process of peasant household entrepreneurial bricolage, thus affecting peasant household entrepreneurial performance.*

## 3. Research Design

### 3.1. Source of Sample Data

The data in this paper came from the research group's research on the entrepreneurial performance of rural households in central Jiangsu Province in December 2022. Farmers in the sample areas have relatively perfect digital infrastructure, superior resource conditions, and high entrepreneurial activity and digital levels in the entrepreneurial process, and from the perspective of regional industrial structure and geographical location, all have a certain representative. The research group adopted the method of stratification and random sampling and selected two to three representative towns in the above counties (cities and districts), randomly selected three to four natural administrative villages in each sample town, and randomly selected 10–12 sample farmers in each sample village and then conducted a one-to-one household questionnaire survey. The subjects of this survey were mainly householders, and the relevant information provided by other family members was also referred to. The survey time of each household was about 30 min. The questionnaire covered basic personal and family information, digital literacy, resource access and opportunity identification, and the entrepreneurial performance of the farmers. In this study, 345 questionnaires were actually sent, and 308 were effectively collected, with an effective rate of 89.28%. The distribution statistics of the survey samples are shown in Table 1.

**Table 1.** Sample distribution statistics (*n* = 308).

| Category | Classification | Sample Number | Proportion (%) | Category | Classification | Sample Number | Proportion (%) |
|---|---|---|---|---|---|---|---|
| Respondent sex | Male | 184 | 60 | Household labor force | 1 | 55 | 18 |
| | Female | 124 | 40 | | 2 | 167 | 54 |
| Respondent age | 36–45 | 148 | 48 | | ≥3 | 86 | 28 |
| | 46–55 | 112 | 36 | Annual household income | ≤50,000 | 36 | 12 |
| | >55 | 48 | 16 | | 60,000–100,000 | 135 | 44 |
| Education level | Primary school and below | 59 | 19 | | 110,000–150,000 | 80 | 26 |
| | Junior high school | 82 | 27 | | 160,000–200,000 | 38 | 12 |
| | Senior high school | 75 | 24 | | >200,000 | 19 | 6 |
| | Vocational college | 54 | 18 | Health status | Very unhealthy | 15 | 5 |
| | University or above | 38 | 12 | | Less healthy | 29 | 9 |
| Years of entrepreneurship | ≤5 | 9 | 2 | | In general | 44 | 14 |
| | 6–10 | 184 | 60 | | Relatively healthy | 119 | 39 |
| | 11–15 | 104 | 34 | | Very healthy | 101 | 33 |
| | >15 | 11 | 4 | | | | |

### 3.2. Measurement of Variables

(1) The dependent variable of this study was the farmers' entrepreneurial performance (ENP). Referring to Lanlan S's scale [15], eight items were selected, and the Likert five-component scale was adopted to measure the farmers' entrepreneurial performance. For specific items, see Table 2. Meanwhile, SPSS 26.0 software was used for factor analysis, and a common factor was extracted by the principal component analysis

method, which was named as the peasant household entrepreneurial performance, and the cumulative variance contribution rate was 59.33%.

(2) The core independent variable of this study was the digital literacy (DL) of farmers. The competencies (C) were the transcendence and integrity of knowledge (K), skills (S), and attitudes (A), i.e., C = (K + S)A [31]. There have been many studies on the definition of digital literacy in academia at home and abroad, but no consensus has been formed, and it cannot meet the characteristics of Chinese farmers under the dual structure of China. Therefore, this paper started from the connotation of literacy, combined the research of Lanlan and Jie [7,10], took into full consideration the production and management reality of farmers, and defined the farmers' digital literacy as the synthesis of relevant digital knowledge, digital ability, and digital consciousness possessed or formed by farmers in production and life practice under the digital context. At the same time, by referring to Honggen's scale [8], digital literacy was divided into three dimensions: digital cognitive identification, digital evaluation application, and digital communication sharing. Seven items were selected, and the Likert five-component scale was adopted to measure the digital literacy of farmers. For specific items, see Table 2.

(3) In this paper, entrepreneurial bricolage and entrepreneurial opportunity identification were selected as the mediating variables. Referring to the maturity scale developed by Senyard et al. [32], the EB included eight questions such as "You will use any existing resources that may be useful to deal with new problems and opportunities". The entrepreneurial opportunity identification (EOI) mainly draws on the mature scale developed by Ozgen and Baron (2007) [33], including four items such as "In your daily life, you can always find entrepreneurial opportunities around you". The Likert five-component scale was used for all intermediary variables. For specific measurement items, see Table 2.

(4) In this paper, relevant variables that may affect entrepreneurial performance were selected as the control variables by referring to existing studies, including the individual characteristics and family characteristics of farmers. Among them, five variables including sex, age, education level, health status, and years of starting a business were selected for individual characteristics of farmers, and two variables including annual household income and the number of labor force members were selected for family characteristics, so as to eliminate the influence of potential factors on the research conclusion as far as possible, in order to obtain more reliable results.

**Table 2.** Reliability test results of measurement items of variables and measurement models.

| Variable | Measurement Item | Load Value | Cronbach's α | CR | AVE |
|---|---|---|---|---|---|
| DL | You think you live in the digital age, and digital information technology is very important to production and life. | 0.710 | 0.903 | 0.904 | 0.577 |
| | When you encounter problems in production and life, you can use digital tools to solve problems. | 0.687 | | | |
| | You have a better understanding of the theories, knowledge, and methods related to digital information. | 0.897 | | | |
| | You can obtain the data and information needed for production and life from mobile phones and other network channels and judge whether it is true or false. | 0.783 | | | |
| | You will use digital tools to solve problems in production and life and make decisions. | 0.779 | | | |
| | You can share your information with others through wechat and other electronic social software. | 0.729 | | | |
| | You are able to consciously follow ethical guidelines in your activities of finding, evaluating, using, and disseminating information. | 0.709 | | | |

**Table 2.** *Cont.*

| Variable | Measurement Item | Load Value | Cronbach's $\alpha$ | CR | AVE |
|---|---|---|---|---|---|
| EOI | In your daily life, you will always find entrepreneurial opportunities around you. | 0.752 | 0.856 | 0.858 | 0.602 |
| | You can effectively identify the products or services that consumers need. | 0.709 | | | |
| | You can quickly choose between opportunities. | 0.819 | | | |
| | You can find new entrepreneurial opportunities in a variety of policies. | 0.817 | | | |
| EB | You use whatever existing resources might be useful to deal with new problems and new opportunities. | 0.916 | 0.894 | 0.897 | 0.527 |
| | You are very confident that you can find a viable solution with the resources available. | 0.771 | | | |
| | You tackle more challenges with your existing resources than any other enterprise. | 0.713 | | | |
| | You address new challenges by integrating existing resources with cheap resources. | 0.622 | | | |
| | When faced with a new problem or opportunity, you may find a workable solution and take action. | 0.734 | | | |
| | By combining existing resources, you address a variety of new challenges. | 0.703 | | | |
| | When faced with a new challenge, you cobble together a workable solution from existing resources. | 0.751 | | | |
| | You combine existing resources that you had planned for other uses to meet new challenges. | 0.537 | | | |
| ENP | The overall operation of your business is good. | 0.699 | 0.901 | 0.902 | 0.536 |
| | Your business is very profitable. | 0.741 | | | |
| | Your business is expanding rapidly. | 0.805 | | | |
| | The market share of your business (sales, business, etc.) is growing rapidly. | 0.698 | | | |
| | You've achieved what you envisioned before you started your business. | 0.647 | | | |
| | Your personal income is much higher than before you started your business. | 0.720 | | | |
| | Your quality of life has greatly improved since you started your business. | 0.790 | | | |
| | After your success, your social status has been greatly improved. | 0.743 | | | |

Note: DL = digital literacy, EOI = entrepreneurial opportunity identification, EB = entrepreneurial bricolage, and ENP = entrepreneurial performance.

## 4. Data Analysis and Hypothesis Testing

### 4.1. Reliability and Validity Test

In this study, SPSS 26.0 software was used to test the reliability of four scales of entrepreneurial farmers' digital literacy, entrepreneurial opportunity identification, entrepreneurial bricolage, and entrepreneurial performance, and the results are shown in Table 2. Cronbach's $\alpha$ values of all variables were greater than 0.8, and CR values were greater than or equal to 0.858, indicating that the internal consistency of the scale and the combined reliability of all variables were good, that is, the questionnaire passed the reliability test. The standardized factor loads were obtained by AMOS 26.0. The load values

were all greater than the critical value of 0.5, and the calculated AVE values were all higher than the threshold value of 0.5, indicating that the cohesive effect of the scale was ideal.

This study used the AMOS26.0 software and selected the $\chi^2/\text{df}$, GFI, NFI, TLI, CFI, and RMSEA index for the key variables such as the confirmatory factor analysis, to distinguish between the validity of the variables and the test model, through comparing the situation of each factor model fitting to verify the validity of the results of the original model. As can be seen from Table 3, the fitting effect of the four-factor model was statistically significantly better than that of other factor combination models, and the fitting effect passed the chi-square test with the best performance, indicating that the concept of the key variables of the four-factor model was clear and easy to distinguish effectively. Therefore, the structural validity and discrimination validity of this model passed the test.

**Table 3.** Summary of fitting indexes for confirmatory factor analysis.

| Number | Model | Factor | $\chi^2/\text{df}$ | GFI | NFI | TLI | CFI | RMSEA |
|--------|-------|--------|------|-----|-----|-----|-----|-------|
| 1 | Four-factor model | DL, EOI, EB, ENP | 1.463 | 0.902 | 0.905 | 0.964 | 0.967 | 0.039 |
| 2 | Three-factor model 1 | DL + EOI, EB, ENP | 3.291 | 0.706 | 0.783 | 0.822 | 0.838 | 0.086 |
| 3 | Three-factor model 2 | DL, EOI, EB + ENP | 2.944 | 0.785 | 0.806 | 0.849 | 0.862 | 0.080 |
| 4 | Two-factor model | DL + EOI, EB + NEP | 4.753 | 0.631 | 0.685 | 0.709 | 0.732 | 0.111 |
| 5 | Single factor model | DL + EOI + EB + ENP | 6.563 | 0.538 | 0.564 | 0.569 | 0.602 | 0.135 |

*4.2. Descriptive Statistics and Correlation Analysis*

In this study, descriptive statistics were used to obtain the mean and standard deviation of the control variables and each principal variable, and a bivariate Pearson correlation analysis was conducted. The specific results are shown in Table 4. There was a significant positive correlation between the key variables, among which digital literacy and entrepreneurial performance showed a significant positive correlation ($r = 0.429$, $p < 0.01$). So far, hypothesis H1 has been preliminarily verified, which provides preliminary support for the regression analysis. The correlation coefficients of six key variables in the model are all smaller than the average variance extracted (AVE) square root of the variable, which provides reliable data support for subsequent hypothesis testing and conforms to the basic theoretical expectations. At the same time, the multicollinearity test of independent variables was conducted in this study, and it was concluded that the value range of the variance inflation factor (VIF) was 1.312–2.626 (VIF < 10), and the value range of the fault tolerance was 0.381–0.762, so the multicollinearity was within the acceptable range.

*4.3. Common Method Deviation Test*

Although the data in this study were obtained through a paired questionnaire survey, there may have been a common method bias due to the fact that all measurement items in the questionnaire were filled in by one person and the influence of environment, understanding, and other factors. Therefore, on the basis of asking entrepreneurial farmers to provide real answers in an anonymous way and removing very professional academic expressions to solve the problem of homology error, this study further conducted a common method deviation test. First, the Harman single-factor test method was used to obtain the variance explanation rate of the first factor of 37%, which was less than the critical standard of 50%. The research results showed that there was no single factor to explain most of the variation. However, due to the simplicity of Harman's single factor test, the reliability of the conclusion was somewhat questioned. Therefore, this study further adopted the "control untested single method latent factor method" to test the common method deviation [34]. Based on the original confirmatory factor model, a method potential factor was added. Comparing the two models revealed that the main fitting coefficient $\chi^2/\text{df}$, GFI, NFI, and RMSEA change of fitting index such as the interval between 0.003 and 0.006 were less than 0.01. Therefore, after adding the common method factor, the change was not obvious, and

the model was not significantly improved, indicating that there was no obvious common method bias.

**Table 4.** Mean values, standard deviations, and correlation coefficients of variables.

| Variable | AVG | SD | 1 | 2 | 3 | 4 | 5 | 6 | 7 | 8 | 9 | 10 | 11 |
|---|---|---|---|---|---|---|---|---|---|---|---|---|---|
| 1 Sex | 0.6 | 0.491 | | | | | | | | | | | |
| 2 Age | 3.68 | 0.730 | −0.130 * | | | | | | | | | | |
| 3 Education level | 3.75 | 1.320 | −0.008 | −0.350 * | | | | | | | | | |
| 4 Health status | 3.85 | 1.126 | 0.056 | −0.135 * | −0.053 | | | | | | | | |
| 5 Years of entrepreneurship | 9.88 | 2.720 | 0.010 | −0.079 | 0.140 * | −0.036 | | | | | | | |
| 6 Number of members of household labor force | 2.11 | 0.688 | 0.099 | −0.054 | 0.008 | −0.042 | −0.085 | | | | | | |
| 7 Annual household income | 11.03 | 5.403 | −0.067 | −0.221 ** | 0.774 ** | −0.047 | 0.126 * | 0.078 | | | | | |
| 8 DL | 3.848 | 0.830 | −0.074 | −0.049 | −0.048 | −0.015 | 0.054 | 0.015 | −0.040 | **0.759** | | | |
| 9 EOI | 3.695 | 0.933 | 0.010 | −0.043 | 0.017 | 0.025 | −0.016 | 0.071 | −0.038 | 0.361 ** | **0.776** | | |
| 11 EB | 3.731 | 0.834 | −0.073 | −0.011 | 0.040 | −0.021 | 0.065 | −0.043 | −0.009 | 0.423 ** | 0.448 ** | **0.726** | |
| 12 ENP | 3.701 | 0.833 | −0.028 | 0.003 | −0.046 | −0.016 | −0.016 | −0.014 | −0.069 | 0.429 ** | 0.495 ** | 0.549 ** | **0.732** |

Note: ** means $p < 0.01$, * means $p < 0.05$ (the same below). $n = 308$. The numbers in bold on the diagonal are the average variance extracted (AVE) values. The sex code 0 represents female, and 1 represents male. The age code 1 represents 25 years and under, 2 represents 26–35 years, 3 represents 36–45 years, 4 represents 46–55 years, and 5 represents 55 years and older. The number of educational attainment (the highest degree) 1 means no schooling, 2 is primary school, 3 is junior high school, 4 is high school, 5 is vocational college, and 6 is university or above. The health code 1 is very unhealthy, 2 is relatively unhealthy, 3 is average, 4 is relatively healthy, and 5 is very healthy. The unit of entrepreneurial life is year. The number of units of household labor force is one. The annual household income refers to the income of a household member in the latest year, and the unit is CNY 10,000. The remaining key variables are on the Likert scale, where 1 represents very inconsistent, 2 represents relatively inconsistent, 3 represents general, 4 represents relatively consistent, and 5 represents very consistent.

### 4.4. Hypothesis Testing

#### 4.4.1. Test of Direct Effect and Intermediate Effect

With the help of SPSS26.0 software, this study used hierarchical regression analysis to examine the influence of digital literacy on the entrepreneurial performance of farmers and the mediating role of entrepreneurial bricolage and entrepreneurial opportunity identification. First, model 1 and model 2 were constructed with entrepreneurial bricolage as the result variables. Model 1 only included control variables (age, sex, education level, health status, years of entrepreneurship, number of family labor force, and annual household income). Model 2 added digital literacy on the basis of model 1, so as to analyze the impact of digital literacy on entrepreneurial bricolage. Second, taking entrepreneurial opportunity identification as the result variable, model 3 only included control variables, and models 4 and 5 added digital literacy and entrepreneurial bricolage on the basis of model 3 to analyze the impacts of digital literacy and entrepreneurial bricolage on entrepreneurial opportunity identification. Finally, entrepreneurial performance was used as the result variable to construct models 6–11, in which model 6 only included control variables. Model 7 was obtained by adding digital literacy to model 6, model 8 was obtained by adding entrepreneurial bricolage, model 9 was obtained by adding entrepreneurial opportunity identification, and model 10 was obtained by adding entrepreneurial opportunity identification, and model 11 was obtained. The direct effect of digital literacy on peasant household entrepreneurial performance and the intermediary effect of entrepreneurial bricolage and entrepreneurial opportunity identification were analyzed. The results of the hierarchical regression are shown in Table 5.

**Table 5.** Results of hierarchical regression analysis of direct effects and intermediate effects.

| Variable | EB | | | EOI | | | ENP | | | | |
|---|---|---|---|---|---|---|---|---|---|---|---|
| | Model 1 | Model 2 | Model 3 | Model 4 | Model 5 | Model 6 | Model 7 | Model 8 | Model 9 | Model 10 | Model 11 |
| Sex | −0.131 | −0.067 | −0.022 | 0.04 | 0.045 | −0.058 | 0.007 | 0.014 | −0.048 | 0.037 | −0.007 |
| Age | −0.005 | 0.037 | −0.035 | 0.006 | −0.033 | −0.022 | 0.021 | −0.019 | −0.006 | 0.004 | 0.019 |
| Education level | 0.072 | 0.093 | 0.083 | 0.103 | 0.046 | 0.008 | 0.029 | −0.032 | −0.029 | −0.013 | −0.007 |
| Health status | −0.012 | −0.004 | 0.02 | 0.027 | 0.026 | −0.015 | −0.008 | −0.009 | −0.024 | −0.006 | −0.018 |
| Years of entrepreneurship | 0.016 | 0.007 | −0.003 | −0.011 | −0.011 | −0.003 | −0.011 | −0.012 | −0.002 | −0.014 | −0.007 |
| Number of members of household labor force | −0.030 | −0.043 | 0.11 | 0.098 | 0.125 | −0.008 | −0.021 | 0.008 | −0.057 | −0.002 | −0.056 |
| Annual household income | −0.017 | −0.016 | −0.024 | −0.023 | −0.016 | −0.013 | −0.012 | −0.004 | −0.002 | −0.005 | −0.004 |
| DL | | 0.426 ** | | 0.411 ** | | | 0.433 ** | | | 0.240 ** | 0.288 ** |
| EB | | | | | 0.506 ** | | | 0552 ** | | 0.451 ** | |
| EOI | | | | | | | | | 0.446 ** | | 0.352 ** |
| $R^2$ | 0.015 | 0.192 | 0.015 | 0.146 | 0.216 | 0.007 | 0.189 | 0.308 | 0.252 | 0.354 | 0.322 |
| $\Delta R^2$ | 0015 | 0.176 | 0.015 | 0.131 | 0.216 | 0.007 | 0.182 | 0.301 | 0.246 | 0.165 | 0.315 |
| F | 0.669 | 8.862 *** | 0.647 | 6.386 *** | 10.301 *** | 0.292 | 8.717 *** | 16.645 *** | 12.610 *** | 18.169 *** | 15.742 *** |

Note: *** means $p < 0.001$, ** means $p < 0.01$.

According to models 2, 4, and 7, rural households' digital literacy had a significant positive effect on entrepreneurial performance ($\beta = 0.433$, $p < 0.01$), entrepreneurial bricolage ($\beta = 0.426$, $p < 0.01$), and entrepreneurial opportunity identification ($\beta = 0.411$, $p < 0.01$), that is, hypotheses H1, H2, and H4 were valid. According to model 8, entrepreneurial bricolage and entrepreneurial opportunity identification had a significant positive effect on entrepreneurial performance ($\beta = 0.552$, $p < 0.01$). By comparing model 7 and model 10, it can be found that after the inclusion of entrepreneurial bricolage, the influence coefficient of rural households' digital literacy on entrepreneurial performance decreased from 0.433 to 0.240, but it was still significant. This indicated that entrepreneurial bricolage played a partial intermediary role in the relationship between rural households' digital literacy and entrepreneurial performance, that is, hypothesis H3 was valid. According to model 9, entrepreneurial opportunity identification had a significant positive effect on the entrepreneurial performance of rural households ($\beta = 0.446$, $p < 0.01$). By comparing model 7 and model 11, it can be found that after adding entrepreneurial opportunity identification, the influence coefficient of rural households' digital literacy on entrepreneurial performance decreased from 0.433 to 0.288, but it was still significant. This indicated that entrepreneurial opportunity identification played a partial intermediary role in the relationship between rural households' digital literacy and entrepreneurial performance, that is, hypothesis H5 was valid.

### 4.4.2. Test of Chain Mediation Effect

The bootstrap method (5000 repeated samples, 95% confidence interval) was used to examine the chain-mediated effects of entrepreneurial bricolage and entrepreneurial opportunity identification on digital literacy and entrepreneurial performance of rural households. The results are shown in Table 6.

As can be seen from Table 6, the total effect value of digital literacy on the entrepreneurial performance of rural households was 0.433, the 95% confidence interval was [0.329, 0.536], the confidence interval did not include 0, and the total effect was significant. The direct effect value of digital literacy on entrepreneurial performance of rural households was 0.182. The indirect effect value of entrepreneurial bricolage was 0.150, and the 95% confidence interval was [0.088, 0.216], which did not contain 0, indicating that entrepreneurial bricolage had a significant mediating effect between digital literacy and rural

household entrepreneurial performance, namely, hypothesis H3 was supported. The indirect effect value of entrepreneurial opportunity identification was 0.058, the 95% confidence interval was [0.019, 0.103], and the confidence interval did not contain 0, which indicated that entrepreneurial opportunity identification had a significant mediating effect between digital literacy and the entrepreneurial performance of farmers, that is, hypothesis H5 was supported. The indirect effect value of entrepreneurial bricolage and entrepreneurial opportunity identification was 0.042, the 95% confidence interval was [0.018, 0.073], and the confidence interval did not contain 0, which indicated that the chain mediation effect was significant, that is, hypothesis H6 was supported.

**Table 6.** Analysis results of Bootstrap chain mediation effect.

| Effect Path | Effect Value | Standard Error | 95% Confidence Interval | |
|---|---|---|---|---|
| | | | Lower Limit | Upper Limit |
| Direct effect | | | | |
| DL → ENP | 0.182 *** | 0.051 | 0.082 | 0.282 |
| Indirect effect | | | | |
| DL → EB → ENP | 0.150 *** | 0.032 | 0.088 | 0.216 |
| DL → EOI → ENP | 0.058 ** | 0.022 | 0.019 | 0.103 |
| DL → EB → EOI → ENP | 0.042 ** | 0.014 | 0.018 | 0.073 |
| Total effect | 0.433 *** | 0.053 | 0.329 | 0.536 |

Note: *** means $p < 0.001$, ** means $p < 0.01$.

## 5. Discussion

First of all, according to the Timmons entrepreneurial factors model theory, entrepreneurial activities cannot be separated from three basic elements: entrepreneur, entrepreneurial opportunity, and entrepreneurial resources. Based on this theory, the entrepreneurial performance of farmers is the result of matching and balancing among these three factors. Second, based on the higher-order theory, when facing the organizational situation, entrepreneurs will have an impact on the resources, opportunities, and strategies of the enterprise according to their personal characteristics within their limited vision, thus affecting the outcome output of the result. Finally, the resource-based theory points out that the availability of resources possessed by enterprises affects the richness of the entrepreneurial opportunity set to a considerable extent. Combined with the higher-order theory, the resource-based theory, and the Timmons entrepreneurial factor model theory, this study analyzed the effect mechanism of rural households' digital literacy on entrepreneurial performance. Based on the theory logic of "resource–opportunity–performance", entrepreneurial bricolage and entrepreneurial opportunity identification were introduced, and a chain intermediary model was constructed. The hierarchical regression analysis and the bootstrap method were used to test the impact of rural households' digital literacy on entrepreneurial performance and the intermediary and chain intermediary effects of entrepreneurial bricolage and entrepreneurial opportunity identification.

Through theoretical analysis and empirical tests, the following conclusions are drawn: (1) Rural households' digital literacy had a significant positive impact on their entrepreneurial performance. This means that the higher the digital literacy of farmers, the more likely to improve the entrepreneurial performance. (2) Digital literacy mediated the entrepreneurial performance of farmers through entrepreneurial bricolage. Digital literacy can promote and help farmers to piece together and reorganize existing resources at hand and increase heterogeneous resources and thus can help farmers to obtain competitive advantages and bring about changes in entrepreneurial performance. (3) Digital literacy mediated the entrepreneurial performance of farmers with entrepreneurial opportunity identification. High digital literacy can help farmers to accurately identify various entrepreneurial opportunities, identify and evaluate these opportunities in a timely manner, and ultimately improve entrepreneurial performance. (4) The effect of digital literacy and the entrepreneurial performance of rural households were not separable from each other. That is, digital

literacy promoted the identification of entrepreneurial opportunities by influencing the entrepreneurial bricolage process of rural households, thus affecting the entrepreneurial performance of the rural households.

## 6. Conclusions

### 6.1. Theoretical Contribution

First of all, most of the existing literature uses "whether" to describe the impact of the accessibility of digital information technology such as the Internet on entrepreneurship. However, with the narrowing of the differences in the accessibility of digital technology in the information age, whether the application ability of digital technology affects entrepreneurship has become the current research focus. From the perspective of digital literacy, this study selected three dimensions of digital cognitive identification, digital evaluation application, and digital communication sharing to accurately quantify the digital literacy of farmers and tested the impact of digital literacy on the entrepreneurial performance of farmers at the micro level, enriching the empirical research on the drive of digital information technology to farmers' entrepreneurship. At the same time, it provided an important supplement to the explanation path of traditional entrepreneurship theory. Second, previous studies have indirectly investigated the positive relationship between digital literacy and entrepreneurial performance, ignoring the specific influence path. This paper explored the influence mechanism of digital literacy on the entrepreneurial performance of farmers from two approaches of entrepreneurial bricolage and entrepreneurial opportunity identification, systematically and comprehensively explained the influence process and internal mechanism of digital literacy on the entrepreneurial performance of farmers, further improved the investigation of the outcome variables of digital literacy, and enriched the research content in the field of digital literacy driving effect. Finally, this paper confirmed that digital literacy can promote the entrepreneurial bricolage and entrepreneurial opportunity identification of rural households, expanded the research on the pre-influencing factors of entrepreneurial bricolage and entrepreneurial opportunity identification under the background of digital economy, and further enriched the theoretical mechanism of entrepreneurial bricolage, entrepreneurial opportunity identification, and digital literacy. In addition, current academic research focuses on factors such as social networks and the entrepreneurial ability of entrepreneurial subjects. This paper discussed the influence of digital literacy on entrepreneurial performance, thus broadening the research on the pre-influencing factors of entrepreneurial performance and providing sufficient theoretical logic and practical guidance for the improvement of entrepreneurial performance of rural households.

### 6.2. Practical Inspiration

(1) Pay attention to the cultivation of rural households' digital literacy, and realize that digital technology enables rural households to start businesses. First of all, on the basis of constantly improving the construction of digital infrastructure, the government has intensified the training of diversified digital skills, such as opening training courses on the application of e-commerce live broadcasting skills, hiring professionals to train and teach, and holding digital skills competitions, so as to improve the digital literacy of farmers and effectively deal with the problems of the farmers' lack of knowledge and skills in the face of new technological scenarios. Lay a solid foundation for farmers to carry out entrepreneurial activities. Second, entrepreneurial farmers should take the initiative to improve their own digital literacy, that is, they should have sensitive digital awareness, rich digital knowledge, and superb digital skills, so as to make full use of the information channel effect and social network effect of digital technology, accurately identify entrepreneurial opportunities, piece together and reorganize resources, and thus better obtain competitive advantages and achieve performance improvement.

(2) Entrepreneurial farmers should seize digital dividends and effectively use digital technologies to improve entrepreneurial performance. By improving digital literacy, entrepreneurial farmers can make better use of digital technology tools to "make do with" and "reconstruct" resources embedded in social networks or redundant resources at hand, so as to achieve orderly and centralized utilization, so as to reduce resource acquisition costs, enhance their own competitive advantages, and improve entrepreneurial performance. At the same time, entrepreneurial opportunity identification is an effective behavior, and digital literacy promotes the improvement of entrepreneurial performance of rural households. Based on the bricolage resources, entrepreneurial farmers can capture a variety of potential business opportunities and timely evaluate and develop entrepreneurial opportunities to improve entrepreneurial performance, so as to promote the sustainable development of rural economy and society.

*6.3. Research limitations and Prospects*

Although this study has certain theoretical and practical significance, there are still some shortcomings: (1) This paper confirmed that entrepreneurial bricolage and entrepreneurial opportunity identification were important intermediate mechanisms for digital literacy to promote the improvement of rural household entrepreneurial performance, but there may be other pathways between digital literacy and rural household entrepreneurial performance. Other mediating effects can be explored in the future, and the moderating effects of other situational factors (such as environmental dynamics) can be considered. (2) This paper took digital literacy as a single dimension and did not further discuss the relationship between different types of digital literacy and their impact on entrepreneurial performance. In the future, we can study the internal relationship between different types of digital literacy and the incremental effects of their interaction on entrepreneurial performance. (3) In terms of sample size, this study mainly focused on specific regions, while sample quantity, inclusiveness, and representativeness need to be expanded. In the future, samples of farmers in different regions and periods can be collected to improve the universality of the conclusion. (4) Cross-sectional data were used in this paper, without considering the influence of time change on the research results. Therefore, longitudinal analysis can be considered in the future to further confirm the robustness of the relationship between variables.

**Author Contributions:** S.J.: designing research and questionnaire survey and writing original draft; J.Z.: conceptualization and funding acquisition. All authors have read and agreed to the published version of the manuscript.

**Funding:** This research was funded by National Social Science Foundation of China (19BGL149).

**Data Availability Statement:** Data are available on request due to restrictions, e.g., privacy or ethical. The data presented in this study are available on request from the corresponding author. The data are not publicly available due to the team research project.

**Acknowledgments:** The author thank the editors and reviewers for their hard work and thank Juan Li and Zenan Sun for their help in revising the paper.

**Conflicts of Interest:** The authors declare no conflict of interest.

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
