# Peer review of "The Impact Path of Digital Literacy on Farmers’ Entrepreneurial Performance: Based on Survey Data in Jiangsu Province"

_sustainability, doi:10.3390/su151411159_

Round 1

Reviewer 1 Report

The paper analyses the impact of digital literacy on the entrepreneurial performance of rural households. I think that the topic is interesting and the paper can be published.

I have some recommendations to the authors in order to improve the paper:

Theoretical framework

The paper contains a reference to   high-level theory and resource-based theory but it is not well explained how this support the analysis and it is not used in the explanations of the results. I recommend to explain the results through the lens of the theoretical framework purposed.

Conclusions

The section is too long and the paper lacks a section of discussion. I recommend that section of conclusion be divided into discussion and conclusions.

Other minor issues:

Table 4 has the title in Chinese

Reviewer 2 Report

The article focuses on farmer's entrepreneurship by discussing the impact of farmer's digital literacy on entrepreneurship. The subject selection makes the study attractive.The paper highlights the importance of the digital literacy on entrepreneurship in the digital era. Second, the paper's theoretical background is well-enough, and it explains with theories. The hypotheses are based on theories, also it is suitable with literature and theory. With these advantages I have doubts and have some suggestions to author(s).    The major problem of the article is the sample size. The number of participants in the study is quite insufficient to generalize the results of the study. Therefore, the sample of the study should be amplified.

On the other hand the minor problems of the article can be detailed as follows. As it can be seen in the attached paper, highlights should be corrected. At the first usage of AVE it should be written in long form, and then abbreviation can be given in brackets. The name of the Table-4 is Chinese, it must be corrected. On Table 2 the variables and measurement items are not understandable, it will be good to group by horizontal line. Cited research in text should be corrected (not Lanlan S and Jie W, it must be Lanlan and Jie; and not SENYARD et al., must be Senyard et al.). References list should be checked according to journal format. Studies written by capital letter must be corrected (Please check: https://www.mdpi.com/journal/sustainability/instructions). Entrepreneurial Bricolage or Entrepreneurial Patchwork, it is bricolage till the section 4.4, and becomes patchwork after this point. So it should be standardized. It would be appropriate to give the explanations about the abbreviations in the tables as a note under the table.
